# Political Hierarchy and Long-Term Effects on TFP: Evidence from a Provincial Elevation in China

Wenjie Hu [ORCID] and Yuguang Yang *

School of Economics, Renmin University of China, Beijing 100872, China
* Correspondence: thomsjr23@163.com

**Abstract:** This paper examines the effects of political hierarchy on firms' long-term total factor productivity using evidence from the elevation of Chongqing to provincial-level government in China. Using the elevation as an exogenous shock and the Sichuan-Chongqing border as a dividing line, we utilized the spatial regression discontinuity approach to identify the causal link. We found that before Chongqing's elevation to a provincial-level municipality, the TFP of Chongqing firms was not significantly different from that of Sichuan firms, but after the elevation, the TFP of Chongqing firms near the border was significantly lower than that of Sichuan firms. The mechanism analysis shows that the land transaction price in Chongqing is significantly lower than that in Sichuan Province, which leads to an abundance of low-productivity firms and the "crowding out" of high-productivity firms. We also found that government intervention in Chongqing exacerbates the degree of resource misallocation at the firm level, which together lead to a decline in firm TFP.

**Keywords:** political hierarchy; TFP; long-run effects; spatial regression discontinuity





## 1. Introduction

Since its political transition in the 1980s, China has gradually moved from a planned economy to a market economy. However, it is undeniable that the economic development of cities in China is still closely related to the influence of administrative power, which the local government can flexibly rely on to intervene in the economy. Administrative power has an obvious spatial hierarchy, which can be reflected in the administrative level of the city [1]. In China, the central government has the highest administrative and economic power, which decreases as the administrative level decreases: municipalities have the same social and economic management power as provincial administrative regions, then prefecture-level cities, and finally county-level cities. Cities at different administrative levels have different economic management powers, political influence, and resource allocation capabilities [2], which not only have important implications for macro-level economic development but also for enterprises.

The development of enterprises is affected by many factors; a large number of studies showed that the total factor productivity (hereinafter referred to as TFP) is the most important factor affecting the sustainable development of enterprises [3]. However, firstly, previous studies mainly focused on the influence of changes happening on specific city levels on enterprise TFP, thus belonging to static analysis; secondly, the existing literature mainly focused on the short-term impact, and few studies examined the impact on enterprises' long-term TFP from administrative-level promotions. The changes in cities' administrative levels can be seen as giving the local government more autonomy; although they are still bound to the central government, the promoted cities have fewer restrictions than before and the officials have much more power in the economic development.

Chongqing has long been part of Sichuan Province. Although it became a separate state-planned city in 1983, it still belonged to Sichuan Province administratively. In 1997, Chongqing separated from Sichuan Province and was elevated from a prefecture-level

city to a provincial-level city with the same administrative powers as Sichuan Province. Therefore, Chongqing's upgrading provides us with a unique sample to study the impact of city-level upgrading on firms' TFP.

First, the new border between Chongqing and Sichuan was an unexpected and exogenous change. Although the demarcation of Chongqing was discussed before 1997, the exact boundary of Chongqing was not announced until it was approved by the central government on 14 March 1997. In fact, the central government had considered four different plans for dividing Sichuan, but it was unclear which plan would be implemented until the central government made a final decision [4]. Second, as mentioned above, under China's unique political system, the decision-making power of local governments depends heavily on their position at the political level. Therefore, the upgrading of Chongqing allowed the newly established municipality to gain substantial powers in economic, administrative and personnel matters. Third, in many cases where new administrative divisions are established, there is usually a significant change in the ethnic composition [5]. In contrast, both in Chongqing and Sichuan, the Han population accounts for around 93% of the total population, and the elevation of Chongqing has not led to significant changes in ethnic composition, so we can rule out the potential impact of changes in ethnic composition. Finally, Chongqing has been part of Sichuan Province for a long time, so it has similar characteristics to Sichuan in terms of culture, system, and social development path.

Taking Chongqing's upgrading to a municipality at the prefecture level in 1997 as an exogenous shock and industrial firms in 2013 as a research sample, we investigate the long-run effects of city-level upgrading on firms' TFP using the spatial regression discontinuity approach. The contributions of this paper are as follows: first, the existing research on the relationship between political hierarchy and TFP mostly focuses on the effect of the existing city level on TFP, which is a static effect, while our paper focuses on the long-term effect of city-level elevation on firms' TFP, which is a dynamic effect. Second, by using Chongqing's elevation as an exogenous shock and the Sichuan-Chongqing border as a dividing line, the spatial regression discontinuity approach can effectively identify the causal relationship between the administrative level and TFP. Moreover, we also analyze the mechanisms involved, which not only helps us to understand the problem but also provides a factual basis for how to solve the problem. Third, the existing literature on city level and TFP studies has mostly focused on short-term effects, but as Krugman stated, productivity is noteverything, but in the long run, it is almost everything [6]. Therefore, this paper provides an important addition and useful extension to the existing literature by showing the long-term effect.

This paper is organized as follows: Section 2 provides the institutional background. Section 3 provides the theoretical analysis. Section 4 explains the research data and identification strategy. Section 5 presents the analysis and discussion of the results. Section 6 reports the mechanism analysis. Section 7 concludes the paper.

## 2. Institutional Background

### 2.1. Institutional Characteristics of the Administrative Level in China

Local governments in China consist of four levels of administrative structure: province–prefecture–county–township. In 2013, there were 34 province-level administrative divisions, 23 provinces, 5 autonomous regions, 4 municipalities subservient to the central government, and 2 special administrative regions. A total of 333 prefecture-level administrative divisions, composed of 286 prefecture-level divisions, 14 regions, 30 autonomous regions, and 3 leagues. In total, there were 2853 county-level administrative units and 40,497 township administrative units. Generally, cities in China do not belong to any specific administrative divisions; they can either refer to province-level, prefecture-level, or county-level administrative divisions. The administrative level of a city is crucial because the decision-making power of Chinese officers is highly dependent on the level of the city [5]. Province-level municipalities have administrative, personnel, and fiscal powers equivalent to those of the province, and prefecture-level cities must obey the policies of the upper provincial

government. Sub-provincial cities, on the other hand, have greater economic and social management powers than ordinary prefecture-level cities, sharing tax revenues directly with the central government, whereas ordinary prefecture-level cities must share tax revenues with the provincial government. However, in terms of administrative divisions, Sub-provincial cities are still subordinated to the upper provincial government.

A peculiarity of China's political hierarchy is that the lower levels of government are completely subordinate to the higher levels of government. From a constitutional point of view, China is a unitary rather than a federal country, and regional powers are vested in the central government. To effectively govern a country with a large population and territory, the central government uses a nested, multi-layered administrative system in which the upper-level government manages and supervises the lower-level government, thereby extending the authority of the central government to all levels of regions [7]. In addition, the unique personnel system of the Chinese central government has further strengthened the dependence of decision-making power on the hierarchical system, in which lower-level officials are evaluated and appointed by the upper-level government. As a result, the upper level of government can easily interfere with the policy decisions of the lower level of government [8]. Moreover, like the administrative system, China's fiscal system is also characterized by a strict vertical hierarchy [9], and the higher-level government has the right to redistribute the fiscal revenue of the lower-level government. Therefore, in China, the elevation of the administrative level not only means the promotion of political status but also the increase in power for administrative, personnel and fiscal affairs.

### 2.2. Chongqing's Elevation to a Provincial-Level Municipality

Historically, Chongqing has long been part of Sichuan Province. Since the Yuan Dynasty (1271–1368 AD), Chongqing has been a region under the jurisdiction of Sichuan Province. In 1996, Sichuan Province consisted of 24 cities or regions, including Chengdu, Chongqing, Fuling, Wanxian and Qianjiang, etc. The original idea of upgrading Chongqing came from a comment by Deng Xiaoping. In early 1985, he suggested that Sichuan was too big to manage, so we could consider dividing it into two parts, one centered on Chengdu and the other on Chongqing. However, the proposal to upgrade Chongqing to a provincial-level municipality made little progress for a long time because of the immature conditions. It was only on 14 March 1997 that the central government decided to merge the former prefecture-level Chongqing with neighboring Fuling, Wanxian and Qianjiang to form Chongqing Municipality, with a total area of 82,000 square kilometers and a population of 30.02 million. The rest of Sichuan province has a population of 85 million.

The establishment of Chongqing Municipality in 1997 was based on three main considerations: First, the central government believed that Chongqing's administrative structure needed to be changed to promote balanced regional economic development and help inland areas catch up with coastal cities. Second, Sichuan had a population of over 100 million and 221 county-level administrative units, including the two megacities of Chengdu and Chongqing. The central government believes that the overpopulation and large administrative area are too costly to manage. Adjusting Sichuan's administrative divisions can simultaneously reduce administrative costs and help Sichuan focus on other areas, especially economic growth in the western minority regions; third, elevating Chongqing can facilitate the construction of the Three Gorges Project and the unified planning, arrangement, and management of migrants in the reservoir area.

The reason we treat Chongqing's elevation as an exogenous shock is that the new border is an unexpected exogenous change. Although the discussion of Chongqing's elevation took place before 1997, only a few top Sichuan officials were involved in the discussion [10]. Moreover, the exact boundary line was not clear until the central government's official announcement. In fact, the central government considered four different alternative demarcation options for the Sichuan–Chongqing border [10].

## 3. Theoretical Analysis

In a unitary state system such as China, the central government has the highest administrative power, which has an obvious spatial hierarchy [1]. As the administrative level decreases, the corresponding administrative power decreases. The central government relies on administrative power to regulate and control the economy. The rank of the administrative level largely determines the amount of political and economic resources, which largely affects the flow and redistribution of resources. Therefore, the upgrading means that Chongqing has gained greater economic and social management authority, and can more flexibly use government intervention to develop the economy, because of which officials can obtain a greater probability of promotion under China's unique political system [8]. At the same time, changes in land management systems have resulted in land playing an increasingly important role in local economic development [11]. Under the current land system, the local government has a monopoly over land supply and can control land prices; to promote industrialization and economic development, the government tends to sell industrial land at low prices and build industrial parks to attract enterprises to settle [12]. Since the land management authority is concentrated in the central and provincial governments [13], Chongqing became autonomous in land management authority after it was separated from Sichuan province.

Based on this point, it can be inferred that raising the administrative level can affect firm TFP in the following two ways: on the one hand, the expansion of land management authority allows the Chongqing government to deliberately lower the land transfer price to attract more firms to settle, which leads to an abundance of many low-productivity firms and the "crowding out" of high-productivity firms, as a result, firm TFP will decline. Many studies used firm-level microdata to confirm the negative impact of low land transfer price on firm TFP [14–16]. The other aspect is that under the political centralization and economic decentralization system in China, the Chongqing government has a stronger incentive to develop the economy after the city's administrative level is upgraded, which will promote the government's policy intervention, which will lead to the misallocation of firm resources and lower firm TFP. Several scholars used empirical tests to support the above view [17,18]. Under the unique development mode of regional competition and promotion of competition in China, land has become an important tool for local governments to attract investment and promote local economic development. In order to gain competitive advantages, local governments often adopt the "race to the bottom" regional competition strategy in the process of attracting investment, and even sell land at a price far lower than the land expropriation cost or even promote "zero land price" to accelerate the process of local industrialization and achieve the goal of economic development. After the elevation of Chongqing, local officials have a stronger incentive to sell land at a lower price, which will largely distort the land transfer price. For enterprises, the low land price not only reduces the one-time prepaid capital of enterprises, alleviating the liquidity constraint during the enterprise's founding period, but the industrial land acquired through the agreement transfer also becomes its assets, and the industrial land can be mortgaged to banks for financing the development of enterprises. Therefore, low land prices may have attracted more companies with poor prospects and excess capacity, dragging down TFP.

## 4. Research Data and Identification Strategy

The data used in this paper are industrial enterprise data from 2013, industrial census data from1995 and land market concession data from 2013, among which the industrial enterprise data are pre-processed by referring to Nie et al. and Chen [19,20]. We restrict the data sample to Sichuan Province and Chongqing; the distribution of industrial enterprises is shown in Figure 1. We can see that, first, the distribution of firms near the center of Chengdu and Chongqing is relatively dense, while the western part of Sichuan province and the eastern part of Chongqing are less dense. Second, many industrial firms are distributed near the border of Sichuan and Chongqing, which is a prerequisite for using the spatial regression discontinuity approach in this paper.

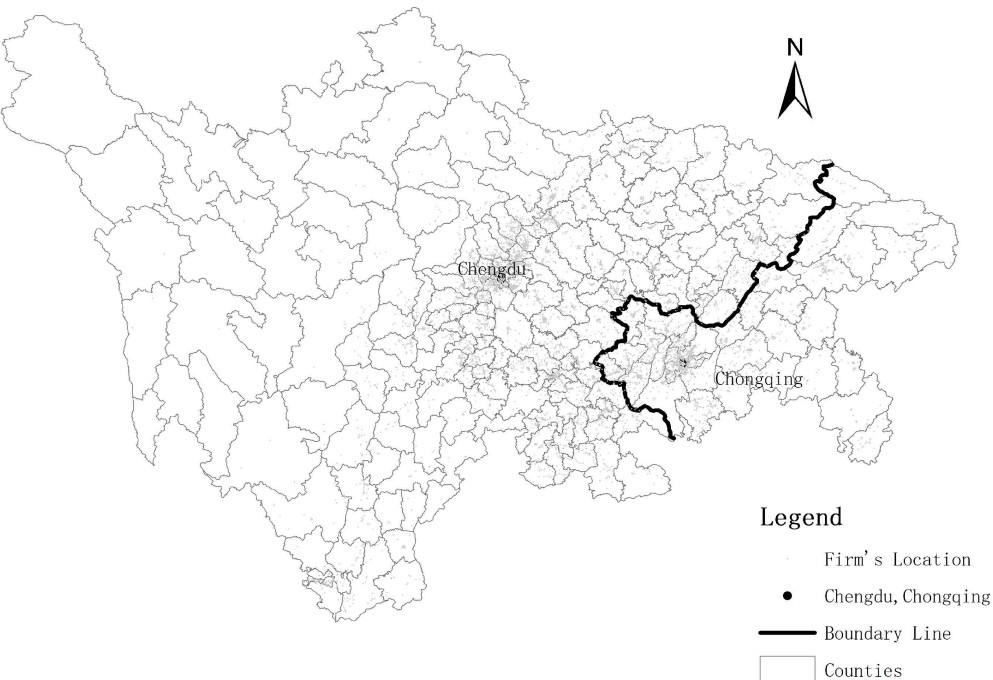

**Figure 1.** Distribution of industrial enterprises in Sichuan Province and Chongqing Municipality.

The explanatory variable in this paper is firm TFP. The methods commonly used in the literature to calculate firm TFP include OLS, FE, OP, and LP, among which OP and LP are considered to be more efficient than OLS [21].We use OLS to calculate TFP for the following reasons: first, the sample of industrial firms in 2013 can only use the OLS method to calculate TFP because of the lack of key indicators when using OP, LP, and ACF methods, such as industrial value added, intermediate goods input. Secondly, the main disadvantage of the OLS method is that it cannot solve the simultaneity deviation and the sample selection deviation. However, fortunately, the existing literature suggests that the TFPs obtained by different methods are not significantly different [22]. Third, the existing literature such as Hsieh and Klenow's study also uses OLS to calculate TFP [23]. Therefore, we use the Solow residual method to calculate firm-level TFP. To check the robustness of the results, we also use labor productivity as an alternative measure of TFP.

Chongqing has long been part of Sichuan Province, and the border between Chongqing and Sichuan Province was only an intra-provincial border until 1997, which ensured that the bordering countries had similar cultural and institutional environments as well as geographic and economic conditions before the shock. This provides us with a very good condition to use the spatial regression discontinuity approach. Therefore, we can use the borders as the discontinuity to conduct a comparative study of firms. The regression has the following form:

$$Y_i = \beta_0 + \beta_1 \text{Chongqing}_i + f(\text{geographic location}_i) + \varepsilon_i \qquad (1)$$

$Y_i$ is the firm-level total factor productivity (TFP), and Chongqing is a dummy variable defined as 1 if the firm is located in Chongqing and 0 otherwise. $f(\text{geographic location}_i)$ is a polynomial controlling geographic location to ensure that the function is smooth at the boundary. Following Dell [24], we use a two-dimensional polynomial in the longitude and latitude of the firm, which can absorb any smoothing tendency in the boundary results. $\beta_1$ is the coefficient we are interested in, indicating the effect of Chongqing's administrative-level elevating on the firm's TFP. $\varepsilon_i$ is the independent and identically distributed error term. Considering that firms in different industries of different counties may be correlated, we use the standard errors of clustering at the county-industry level.

As discussed by Gelman and Imbens [25], Equation (1) can be estimated using two methods: the nonparametric local linear regression or the global polynomial regression. In our benchmark regression, we used local linear regression, which uses a narrow bandwidth near the boundary and controls for a linear latitude–longitude polynomial. As there is no widely accepted two-dimensional optimal bandwidth [26], we restricted our sample to firms within 30 km of the border (Figure 2). We also consider bandwidths of 10, 20 and 50 km to ensure that our estimates are robust to a particular choice of bandwidth. Finally, to test the robustness of the results, we report results using a global multidimensional regression.

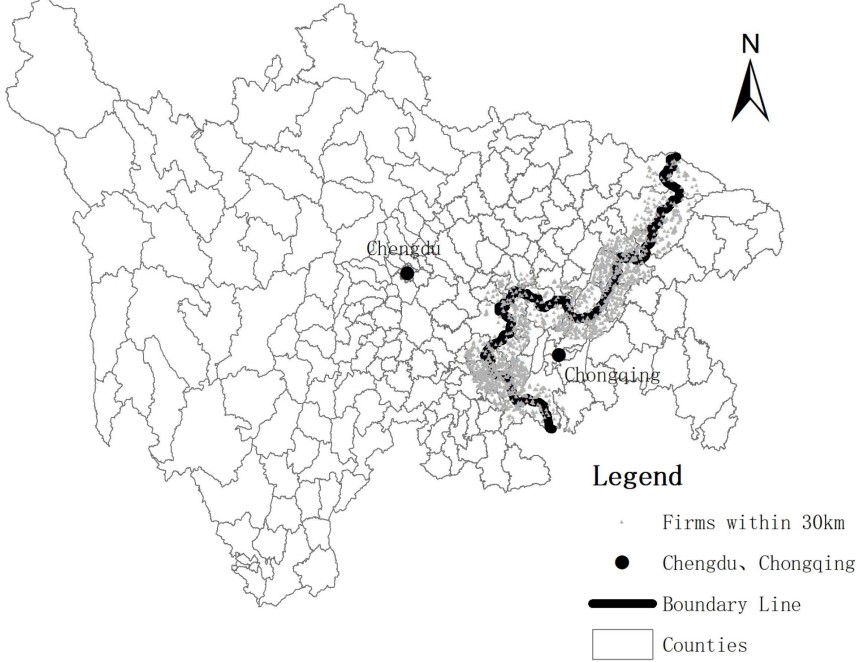

**Figure 2.** Distribution of enterprises located within 30 km of the Sichuan–Chongqing border.

Table 1 shows the t-test results of the main variables for the counties within 30 km of the Chongqing–Sichuan border. It shows that the counties near the border do not differ significantly in slope and altitude, so they can be considered as smoothed. The same applies to the average annual temperature. The data are from meteorological stations in 2013. As complete climate data at the county level were not available, we use the data from the weather stations to calculate the climate data for each county, following the method of Shephard [27]. Specifically, the distance between the county and each weather station is used in the construction of the weights, i.e., the meteorological characteristics of a county should be more like the closer weather station. The weight of each weather station is constructed as follows:

$$W_i = \frac{dist_i^{-2}}{\sum_j dist_j^{-2}}$$

$W_i$ represents the weight of the weather station, and $dist_i$ represents the spherical great circle distance between the weather station $i$ and county $c$. We calculated the distances from the border counties to downtown Chengdu and downtown Chongqing, where the local government is located. The results show that there is no significant difference in the distance from the border counties to Chengdu, while there is only a difference at the 10% significance level to Chongqing. Overall, we can conclude that there is no significant difference in the geographical characteristics of the counties on the border between Sichuan and Chongqing.

**Table 1.** *t*-test of main variables for counties within 30 km of the Chongqing-Sichuan border.

| Variable | Sichuan | | Chongqing | | Variance of Average |
|---|---|---|---|---|---|
| | Sample | Average | Sample | Average | |
| Gradient | 28 | 0.546 | 24 | 0.880 | −0.334 |
| Altitude | 23 | 371.783 | 20 | 342.150 | 29.633 |
| Average temperature | 28 | 17.777 | 24 | 18.047 | −0.269 |
| Distance to Chengdu (KM) | 28 | 286.592 | 24 | 293.328 | −6.736 |
| Distance to Chongqing (KM) | 28 | 197.957 | 24 | 91.505 | 106.452 |
| Log(GDP) | 28 | 2.871 | 24 | 3.099 | −0.228 |
| Log(GDP per capita) | 28 | 7.837 | 24 | 7.984 | −0.147 |
| Log(industrial output per capita) | 28 | 7.591 | 24 | 7.743 | −0.152 |
| Urbanization rate | 28 | 16.126 | 24 | 15.179 | 0.947 |
| Proportion of manufacturing population | 28 | 5.142 | 24 | 6.629 | −1.487 |
| Proportion of ethnic minorities | 28 | 0.245 | 24 | 0.432 | −0.187 |

Given that macroeconomics has an important impact on the development of enterprises, we also examine the economic characteristics of border counties. The GDP, GDP per capita, gross industrial output value per capita and urbanization rate are obtained from the statistical yearbooks of Sichuan Province and Chongqing Municipality in 1996. The GDP, GDP per capita and gross industrial output value per capita of the counties in Chongqing are all slightly better than those of Sichuan Province, but the urbanization rate is slightly lower than that of Sichuan. Importantly, none of the *t*-test results are significant, indicating that there is no significant difference in the economic characteristics of the border counties before Chongqing's elevation. We also compare the share of manufacturing and the share of ethnic minorities in the border counties with data from the 2000 Chinese Census, and the results are still not significantly different. Finally, we examine the TFP of firms located 30 km from the border between Chongqing and the rest of Sichuan Province using 1995 Census data. Figure 3 shows that there is no significant difference in the TFP of industrial enterprises located on either side of the border line in 1995, and the coefficients in Table 2 support this conclusion. The above results show that there is no significant initial economic and industrial difference between the counties near the border, i.e., the conditions for using the spatial regression discontinuity approach are met.

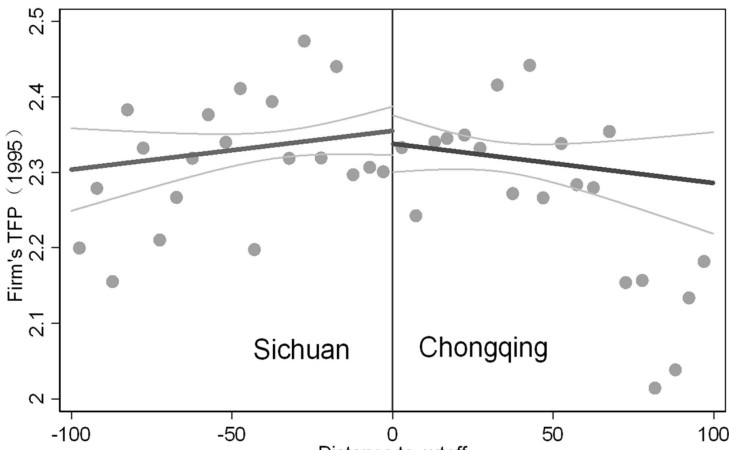

**Figure 3.** TFP of industrial enterprises in Sichuan Province and Chongqing Municipality in 1995. Note: The x-axis represents the distance of the firm from the Chongqing–Sichuan border, and a negative number represents the control group (Sichuan). The gray dots show the average TFP for firms located 5 km from the border. The bold line is a linear fit line using triangular weights developed by Calonico et al. [28,29], i.e., the closer the firm is to the border, the higher the weight, and the thin gray curve shows the 90% confidence interval. The way the weight matrix is constructed is explained in Appendix A.

**Table 2.** Results of industrial enterprises in 1995.

| | (1) TFP | (2) TFP | (3) TFP | (4) TFP | (5) TFP | (6) TFP |
|---|---|---|---|---|---|---|
| | Local Linear | | Local Polynomial | | Global Polynomial | |
| | <30 km | <50 km | <30 km | <50 km | Full Sample | Full Sample |
| Chongqing | −0.0287 | −0.0133 | −0.0589 | −0.0512 | −0.0191 | −0.0088 |
| | (0.043) | (0.035) | (0.042) | (0.035) | (0.025) | (0.025) |
| Polynomial | linear | linear | quadratic polynomials | quadratic dimensional | cubic dimensional | biquadratic dimensional |
| Average TFP | 2.34 | 2.34 | 2.34 | 2.34 | 2.32 | 2.32 |
| N | 3679 | 6063 | 3679 | 6063 | 23,346 | 23,346 |
| Adj. R² | 0.0000 | 0.0008 | 0.0081 | 0.0071 | 0.0040 | 0.0068 |

Note: All regressions control for two-dimensional geographic variables. Values in parentheses are clustering robust standard errors, all regression models are adjusted for clustering at the county level.

## 5. Results and Discussion

### 5.1. Benchmark Results

Table 3 shows the results of the benchmark regressions using the spatial regression discontinuity approach. Column (1) shows the results using the local linear method, where the distance to the border is limited to 30 km. The coefficient of the core explanatory variable we are interested in is −0.27, which is significantly negative at the 1% level. Column (2) shows the results using the same method but limiting the distance to 50 km, which is like the result in column (1). Columns (3) and (4) replace the linear polynomials in latitude and longitude with quadratic polynomials with bandwidths of 30 km and 50 km respectively. Columns (5) and (6) report the results of the global polynomial approach using the full Sichuan and Chongqing samples and controlling for cubic and biquadratic polynomials, respectively. In the above results, the coefficients of the core explanatory variables are all significantly negative at the 1% level, and the coefficients are basically stable at around −0.30. This result indicates that the upgrading of Chongqing to a provincial-level municipality has a significant negative effect on the TFP of firms, with a magnitude ranging from 5.5% to 7.4%.

**Table 3.** Results for industrial enterprises in 2013.

| | (1) TFP | (2) TFP | (3) TFP | (4) TFP | (5) TFP | (6) TFP |
|---|---|---|---|---|---|---|
| | Local Linear | | Local Polynomial | | Global Polynomial | |
| | <30 km | <50 km | <30 km | <50 km | Full Sample | Full Sample |
| Chongqing | −0.2695 *** | −0.2662 *** | −0.3617 *** | −0.3569 *** | −0.3425 *** | −0.3408 *** |
| | (0.048) | (0.044) | (0.051) | (0.051) | (0.042) | (0.042) |
| Conley s.e. | [0.090] | [0.088] | [0.086] | [0.111] | [0.108] | [0.132] |
| Polynomial | linear | linear | quadratic polynomials | quadratic dimensional | cubic dimensional | biquadratic dimensional |
| Average TFP | 4.95 | 4.89 | 4.95 | 4.89 | 4.72 | 4.72 |
| N | 3206 | 5594 | 3206 | 5594 | 17,239 | 17,239 |
| Adj.R² | 0.0444 | 0.0342 | 0.0699 | 0.0442 | 0.0543 | 0.0543 |

Note: *** represent 1% levels of significance. All regressions control for two-dimensional geographic variables. Values in parentheses are clustering robust standard errors, all regression models are adjusted for clustering at the county level.

To account for the possibility of spatial correlation, we report in square parentheses the two-dimensional spatial standard error of Conley [30], where the critical value is set at 1 degree of latitude and 1 degree of longitude, i.e., two firms are assumed to be correlated if their geographical distance is within 1 degree of longitude and 1 degree of latitude (1 degree of latitude is approximately 111 km). We find that the benchmark results still hold; the results are shown in Figure 4, from which we can see that there is a clear discontinuity at

the border between Sichuan Province and Chongqing, i.e., the TFP of firms located within the border of Chongqing is significantly lower than that of Sichuan, which is consistent with the benchmark results in Table 3.

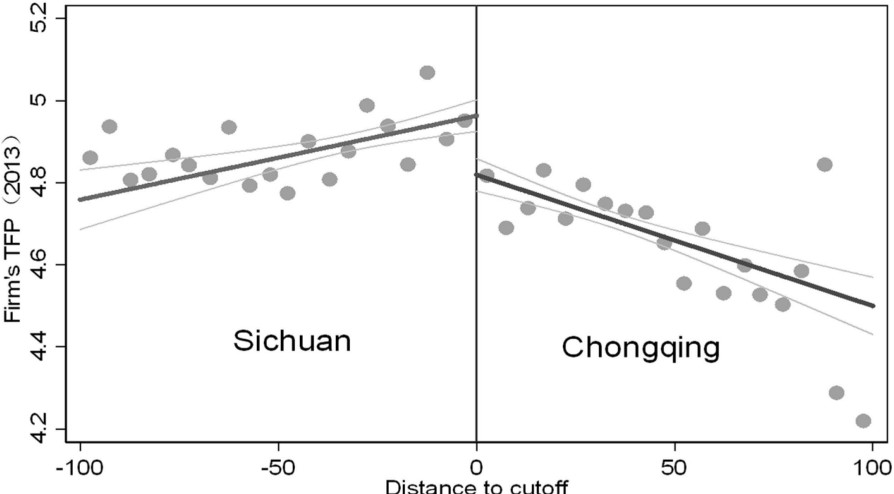

**Figure 4.** TFP of industrial enterprises in Sichuan Province and Chongqing Municipality in 2013. The results show a significant difference in the TFP of firms located on either side of the border. Specifically, firms in Chongqing Municipality have significantly lower TFP than their counterparts in Sichuan Province, which serves as the control group.

*5.2. Robustness Tests*

5.2.1. DID Regression Discontinuity

In this section, we report the results using double difference-in-difference (DID) with regression discontinuity. The regression discontinuity method (RD) can be used to identify local effects near natural experiments or structural policy changes, while the DID (difference in differences) can be used to identify causality based on individual time dimension difference construction. In our paper, these two methods were combined to identify the policy effects of allowing other covariables outside the core explanatory variable (Sichuan–Chongqing boundary) to appear before and after the reform date by using the differences between the two dimensions of longitude and latitude. At the same time, in DID estimation, we control the firm fixed effect and the time fixed effect. The advantages are as follows: Firstly, controlling the fixed effect at the firm level can capture the time stationary individual characteristics that affect TFP at the firm level and further eliminate the influence of unobserved factors. Secondly, this paper mainly studies the long-term impact of policies under Chongqing Municipality on enterprises' TFP. By controlling the fixed time effect, we captured the impact of macro-impact from the national level on the results, which better identifies the long-term impact of the city-level promotion.

Specifically, we use a sample of firms located near the Sichuan–Chongqing border in both 1995 and 2013 to estimate the long-run impact of Chongqing's elevation on firms' TFP. Since the interval between 1995 and 2013 is too long and the average life span of Chinese firms is relatively short, the sample can only be limited to the vicinity of the Sichuan–Chongqing border, therefore, special attention must be paid to firm matching. We matched industrial firms in both 1995 and 2013, and obtained 837 samples, accounting for only 1.56% of the total 1995 sample, while only 100 firms were located within 30 km of the border, accounting for 11.95% of the total matched sample. Therefore, the sample size in this section is very small compared to the benchmark results. The regression results are presented in Table 4.

**Table 4.** DID regression discontinuity.

| | (1) TFP <30 km | (2) TFP <50 km | (3) TFP <100 km | (4) TFP_Labor <30 km | (5) TFP_Labor <50 km | (6) TFP_Labor <100 km |
|---|---|---|---|---|---|---|
| Chongqing * Post | −0.1683 | −0.4639 *** | −0.6213 *** | −0.2284 | −0.4641 | −0.6319 *** |
| | (0.201) | (0.154) | (0.121) | (0.479) | (0.307) | (0.232) |
| firm fixed effect | Y | Y | Y | Y | Y | Y |
| time fixed effect | Y | Y | Y | Y | Y | Y |
| N | 154 | 310 | 632 | 156 | 312 | 634 |
| Adj. $R^2$ | 0.7435 | 0.7196 | 0.7009 | 0.2299 | 0.2019 | 0.1863 |

Note: * and *** represent 10% and 1% levels of significance respectively. All regressions control for two-dimensional geographic variables. Values in parentheses are clustering robust standard errors, all regression models are adjusted for clustering at the county level.

Column (1) shows the results for the samples within 30 km; the sample size is only 154, and the coefficients of the core explanatory variables are not significant, but, fortunately, the absolute values of the coefficients are still not small, indicating that the insignificant results may be caused by the small sample size. Column (2) shows the test results for the samples within 50 km; the sample size has increased, and the coefficients of the core explanatory variables are significantly negative, consistent with the results of the benchmark regression. Column (3) shows the estimation results for the samples within 100 km, which are still significantly negative. In columns (4)–(6), we use labor productivity as a robustness test and the results remain consistent with the first three columns.

5.2.2. Substitution of the Dependent Variable

Considering that the TFP obtained by OLS may be biased because it does not solve the endogenous problem, we use labor productivity as an explanatory variable for robustness testing, the results of which are presented in Table 5. The coefficients of the main explanatory variables remain significantly negative at the 1% level. Taking column (1) as an example, Chongqing's upgrading to a provincial-level municipality reduces the labor productivity of firms by about 27.9%, which is about 2.2% of the average labor productivity. The result is smaller than the benchmark, mainly because labor productivity does not take capital into account, while the TFP used in Table 3 takes both labor and capital into account. This result confirms the robustness of the benchmark results.

**Table 5.** Labor productivity of industrial enterprises.

| | (1) TFP_Labor Local Linear <30 km | (2) TFP_Labor Local Linear <50 km | (3) TFP_Labor Local Polynomial <30 km | (4) TFP_Labor Local Polynomial <50 km | (5) TFP_Labor Global Polynomial Full Sample | (6) TFP_Labor Global Polynomial Full Sample |
|---|---|---|---|---|---|---|
| Chongqing | −0.2790 *** | −0.2229 *** | −0.3328 *** | −0.3364 *** | −0.3419 *** | −0.3405 *** |
| | (0.067) | (0.061) | (0.077) | (0.072) | (0.056) | (0.056) |
| Polynomial | linear | linear | quadratic polynomials | quadratic dimensional | cubic dimensional | biquadratic dimensional |
| Average TFP_labor | 12.87 | 12.89 | 12.87 | 12.89 | 12.87 | 12.87 |
| N | 3206 | 5595 | 3206 | 5595 | 17,241 | 17,241 |
| Adj.$R^2$ | 0.0254 | 0.0143 | 0.0319 | 0.0251 | 0.0173 | 0.0174 |

Note: *** represents 1% levels of significance. All regressions control for two-dimensional geographic variables. Values in parentheses are clustering robust standard errors, all regression models are adjusted for clustering at the county level.

5.2.3. Controlling for Fixed Effects

In Section 4, we provide that there are no differences in the economic and industrial status of border counties prior to Chongqing's elevation, but that there may be industry differences at the firm level. The existence of differences in industry TFPs would result in

this paper capturing effects that do not belong to city-level upgrading. Therefore, in this part of the paper, we add control variables to further control for the industry fixed effects of firms as well as firm ownership.

Specifically, we control for the number of employees, the size of the firm, the ROA and the debt-equity ratio. The results are reported in Table 6. The coefficients of the core explanatory variables are still significantly negative and larger in absolute value after controlling for industry fixed effects. For example, the coefficient in column (3) is about 8.4% lower than the average. The corresponding result in column (3) of Table 3 shows that the coefficient is about 7.4% lower than the average, with no significant change in the results. This suggests that the benchmark results do not capture the effects of TFP due to industry differences.

**Table 6.** Regression results after adding control variables and controlling for industry fixed effects.

| | (1) TFP | (2) TFP | (3) TFP | (4) TFP | (5) TFP | (6) TFP |
|---|---|---|---|---|---|---|
| | **Local Linear** | | **Local Polynomial** | | **Global Polynomial** | |
| | **<30 km** | **<50 km** | **<30 km** | **<50 km** | **Full Sample** | **Full Sample** |
| Chongqing | −0.3455 *** | −0.3219 *** | −0.4170 *** | −0.3948 *** | −0.3536 *** | −0.3502 *** |
| | (0.044) | (0.040) | (0.047) | (0.046) | (0.038) | (0.038) |
| Polynomial | linear | linear | quadratic polynomials | quadratic dimensional | cubic dimensional | biquadratic dimensional |
| Control variables | Y | Y | Y | Y | Y | Y |
| Industry fixed effects | Y | Y | Y | Y | Y | Y |
| Ownership fixed effects | Y | Y | Y | Y | Y | Y |
| Average TFP | 4.95 | 4.89 | 4.95 | 4.89 | 4.72 | 4.72 |
| N | 2993 | 5249 | 2993 | 5249 | 16,392 | 16,392 |
| Adj.$R^2$ | 0.2086 | 0.1801 | 0.2254 | 0.1861 | 0.1878 | 0.1878 |

Note: *** represents 1% levels of significance respectively. All regressions control for two-dimensional geographic variables. Values in parentheses are clustering robust standard errors, all regression models are adjusted for clustering at the county level.

### 5.2.4. Bandwidth and Boundary Adjustment

We limited the sample to 30 and 50 km on either side of the boundary in the benchmark results. To check that the results are not driven by the bandwidth setting, we further limit the boundary distance to 10 and 20 km, the results are shown in columns (1)–(4) of Table 7. The coefficients of the core explanatory variables are still significantly negative, and the magnitudes of the coefficients resemble the benchmark results, indicating that the benchmark results in this paper are not caused by the choice of a particular bandwidth.

When using the spatial regression discontinuity approach for causal identification, we need to ensure that, except for the shocks caused by the border, all other factors are stable on both sides of the border. Although we have confirmed that most of the observable geographic and economic characteristics do not differ significantly, we cannot rule out the existence of other confounding factors that would cause the conditions for using the spatial regression discontinuity approach to be unsatisfied. Therefore, to further eliminate these factors, we will continue the regression analysis by "moving" the real border between Sichuan and Chongqing. Specifically, we made two "moves".

First, we move the boundary line between Sichuan and Chongqing 30 km to the east and construct the treatment and control groups with firms within 30 km of the shifted boundary line. In this case, if there are other unobservable factors affecting the firm's TFP, we should not observe any differences between the two groups. The results are presented in column (5) of Table 7. We can see that the coefficient is negative but insignificant. More importantly, the coefficient is particularly small compared to the benchmark results, indicating that there are no significant differences at the artificial borders. Secondly, we move the boundary line 30 km to the west; column (6) shows the results. We find that

although the coefficient at this point is significantly positive at the 10% significance level, the absolute value of the coefficient is still very small.

**Table 7.** Results after band and boundary adjustment.

| | (1) TFP | (2) TFP | (3) TFP | (4) TFP | (5) TFP | (6) TFP |
|---|---|---|---|---|---|---|
| | Local Linear | | Local Polynomial | | Global Polynomial | |
| | <10 km | <20 km | <10 km | <20 km | 30 km | 30 km |
| Chongqing | −0.2043 ** | −0.2783 *** | −0.3220 *** | −0.3876 *** | | |
| | (0.091) | (0.060) | (0.090) | (0.061) | | |
| Chongqing_east | | | | | −0.0372 | |
| | | | | | (0.035) | |
| Chongqing_west | | | | | | 0.0614 * |
| | | | | | | (0.034) |
| Polynomial | linear | linear | quadratic polynomials | quadratic dimensional | cubic dimensional | biquadratic dimensional |
| Average TFP | 4.89 | 4.94 | 4.89 | 4.94 | 4.95 | 4.95 |
| N | 662 | 1730 | 662 | 1730 | 3378 | 3572 |
| Adj.R² | 0.0563 | 0.0484 | 0.1250 | 0.0880 | 0.0493 | 0.0072 |

Note: *, **, and *** represent 10%, 5%, and 1% levels of significance respectively. All regressions control for two-dimensional geographic variables. Values in parentheses are clustering robust standard errors, all regression models are adjusted for clustering at the county level.

### 5.2.5. Placebo Test

To further eliminate the influence of unobservable factors, we use randomly constructed artificial boundaries of Sichuan and Chongqing for validation. Specifically, we refer to the methods of Oto and Romero and Jia et al. to construct random artificial boundary lines of Sichuan and Chongqing (non-linear, see Figure 5) [31,32], then generate artificial treatment and control groups based on the artificial boundary lines for regression, repeat this process 1000 times, and compare the coefficients with the real coefficients. Figure 6 shows the cumulative distribution of the coefficients obtained from the 1000 iterations. More than 90% of the placebo coefficients are greater than the "true" benchmark regression coefficient (−0.27), indicating that the results of this paper are not coincidental, but caused by the elevation of Chongqing to a provincial-level municipality.

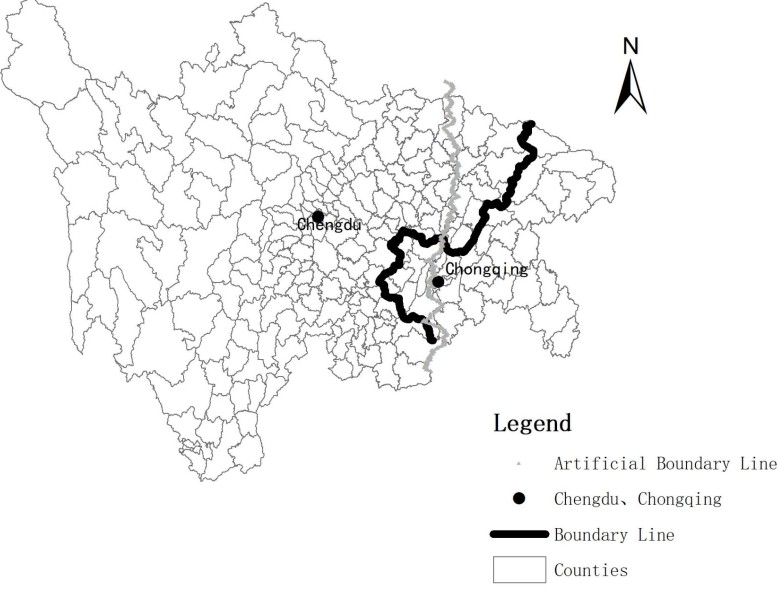

**Figure 5.** A randomly constructed placebo boundary line.

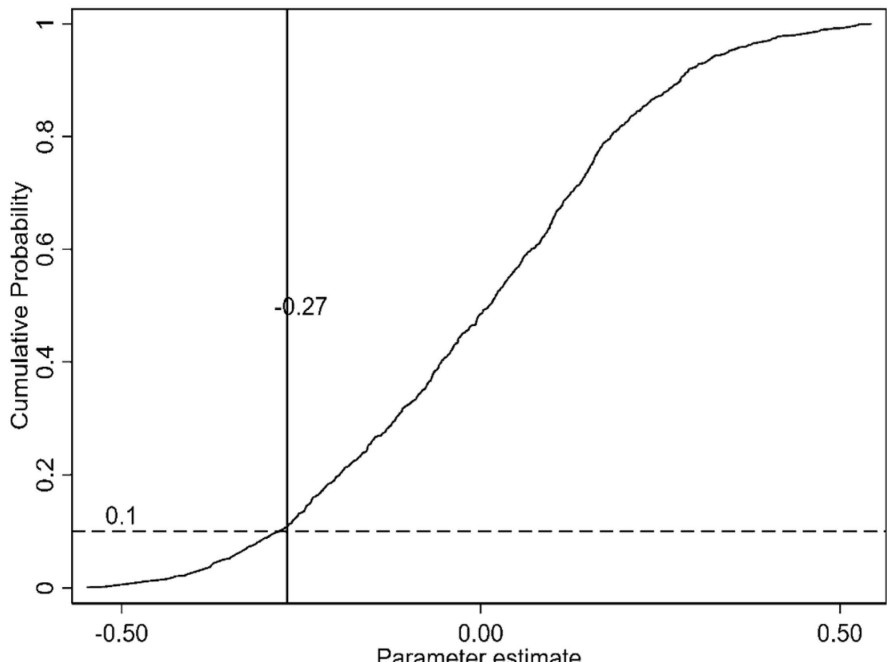

**Figure 6.** Cumulative distribution function of placebo coefficient.

5.2.6. Eliminating the Effect of Firm Migration

In the previous empirical tests, we use firms located near the border in Sichuan Province as a control group for firms in Chongqing, mainly because the geographical and economic characteristics are smoothed near the Sichuan–Chongqing border. However, a major concern is that Chongqing's upgrading may affect firms in Sichuan; for example, some firms may have moved from Sichuan to Chongqing after Chongqing's upgrading, and Sichuan Province may move firms previously located in Chongqing to Sichuan Province before Chongqing's upgrading. In both cases, our previous tests only obtained the average treatment effect in Chongqing's upgrading, which means that the impact of the above situation is also included.

In this part, we eliminate the impact of the above-mentioned situation on firms' TFP mainly by two methods: first, we use the 1995 industrial census data to match the 1998 industrial data, and calculate the transfer rate of firms based on the matching results; we found that there is almost no transfer of firms, and the results are similar to Jia et al. [32].We follow the approach of Jia et al. and Ehrlich and Seide [32,33], and exclude the sample of firms 5 and 10 km away from the border. In general, transfer costs for firms and workers increase with distance, so the migration effect decreases with increasing distance from the Sichuan–Chongqing border. Therefore, excluding the firms 5 and 10 km from the border can eliminate the impact of the migrated firm on the results to the greatest extent.

The regression results are shown in Table 8. Compared to the benchmark results in column (1), Chongqing's altitude has a stronger negative effect on firms' TFP when firms 5 and 10 km from the border are excluded. The results in columns (4)–(6) are similar. This suggests that the results in this paper are not affected by firm migration. In summary, we conclude that the elevation of Chongqing to a provincial-level municipality has a negative effect on firm TFP.

**Table 8.** Results excluding the impact of enterprise migration.

| | (1)<br>TFP<br><br>Benchmark | (2)<br>TFP<br><30 km<br>Excluding 10<br>km | (3)<br>TFP<br><br>Excluding 5<br>km | (4)<br>TFP<br><br>Benchmark | (5)<br>TFP<br><50 km<br>Excluding 10<br>km | (6)<br>TFP<br><br>Excluding 5<br>km |
|---|---|---|---|---|---|---|
| Chongqing | −0.2695 *** | −0.3155 *** | −0.3002 *** | −0.2662 *** | −0.3044 *** | −0.2926 *** |
| | (0.048) | (0.049) | (0.049) | (0.044) | (0.046) | (0.045) |
| Polynomial | linear | linear | linear | linear | linear | linear |
| Average TFP | 4.95 | 4.96 | 4.95 | 4.89 | 4.89 | 4.88 |
| N | 3206 | 2544 | 2914 | 5594 | 4932 | 5302 |
| Adj.R² | 0.0444 | 0.0475 | 0.0461 | 0.0342 | 0.0362 | 0.0356 |

Note: *** represent 1% levels of significance respectively. All regressions control for two-dimensional geographic variables. Values in parentheses are clustering robust standard errors, all regression models are adjusted for clustering at the county level.

## 6. Analysis of Mechanism

### 6.1. Land Price

We use land transactions in Sichuan Province and Chongqing Municipality in 2013 as the sample, where the land price is equal to the price of land/area of land and take the logarithm. Similarly, we limit the distance to the border to 30 and 50 km, control for the fixed effects of land and industry, and use robust standard errors clustered at the county-firm level. The results are shown in Table 9. The coefficients of the core explanatory variables are all negative, indicating that land prices in Chongqing are on average significantly 25% lower than in Sichuan.

**Table 9.** Regression results for land price.

| | (1)<br>Price<br>Local Linear<br><30 km | (2)<br>Price<br><br><50 km | (3)<br>Price<br>Local Polynomial<br><30 km | (4)<br>Price<br><br><50 km | (5)<br>Price<br>Global Polynomial<br>Full Sample | (6)<br>Price<br><br>Full Sample |
|---|---|---|---|---|---|---|
| Chongqing | −0.1639 | −0.2508 ** | −0.2551 | −0.3046 ** | −0.1927 ** | −0.2489 *** |
| | (0.189) | (0.121) | (0.210) | (0.125) | (0.082) | (0.084) |
| Polynomial | linear | linear | quadratic<br>polynomials | quadratic<br>dimensional | cubic<br>dimensional | biquadratic<br>dimensional |
| Land usage | Y | Y | Y | Y | Y | Y |
| Industry | Y | Y | Y | Y | Y | Y |
| N | 1920 | 4595 | 1920 | 4595 | 16,602 | 16,602 |
| Adj.R² | 0.3784 | 0.3660 | 0.3782 | 0.3665 | 0.3589 | 0.3598 |

Note: **, and *** represent 5%, and 1% levels of significance respectively. All regressions control for two-dimensional geographic variables. Values in parentheses are clustering robust standard errors, all regression models are adjusted for clustering at the county level.

Next, we examine the relationship between land price and firm TFP. According to our theoretical analysis, low land prices attract the entry of low-productivity firms because land is finite, which prevents the entry of high-productivity firms, leading to a crowding-out effect. Since land transaction data and firm data cannot be precisely matched, only the average land price at the county level is matched with firm data. The results show that the coefficient of land price is significantly positive, which means that the higher the land price, the higher the TFP. According to the results in Table 9, the land price in Chongqing is significantly lower than that in Sichuan Province, indicating that the land price in Chongqing has a significant impact on the firm's TFP, which is consistent with our theoretical analysis.

### 6.2. Resource Misallocation

In this part, we test whether the administrative intervention after Chongqing's evaluation has exacerbated firms' resource misallocation. We refer to the method of Jiang et al. to calculate two measures of resource misallocation [2]. One method is to calculate the standard deviation of TFP for the dichotomous industry in which each firm is located, and then to obtain the standard deviation of TFP for each firm by taking the weighted average of the share of each firm's operating revenue in the total operating revenue of the industry. The economic implication of this indicator is that if there is no distortion in the allocation of resources, factors of production will flow freely from low-productivity firms to high-productivity firms, and high-productivity firms will eventually merge with low-productivity firms or drive them out of the market. Ideally, the TFP of all firms should converge, and the greater the difference in TFP between firms, i.e., the higher the value of the indicator, the more severe the distortion in resource allocation [2,34].

On the other hand, we use the comprehensive corporate tax rate to measure the extent of resource allocation distortions [35]. Unlike product and factor price distortions, this measure mainly captures allocation distortions among firms in the same industry due to different government interventions such as taxes and subsidies. It is calculated as follows: first, calculate the comprehensive tax rate of each firm, i.e., (tax−subsidy)/revenue. Second, calculate the comprehensive tax rate of the industry in which the firm is located, i.e., (total industry tax−total industry subsidy)/total industry sales, thus obtaining the ratio of the comprehensive tax rate of the firm to the industry. Third, the standard deviation of the ratio is calculated, and the weighted average value is obtained according to the proportion of the firm's operating income to the total operating income of the industry.

The results of the regressions are presented in Tables 10 and 11. Table 10 shows resource misallocation as measured by the standard deviation of TFP, while Table 11 shows resource misallocation as measured by the standard deviation of the comprehensive corporate tax rate. The coefficients of the core explanatory variables are both significantly positive regardless of which measure is used and regardless of the restricted distance to the border as well as the polynomial in latitude and longitude, indicating that Chongqing's elevation to a provincial-level municipality exacerbates the degree of resource misallocation of firms.

**Table 10.** Results of resource misallocation measured by the standard deviation of TFP.

| | (1) Misallocation | (2) Misallocation | (3) Misallocation | (4) Misallocation | (5) Misallocation | (6) Misallocation |
|---|---|---|---|---|---|---|
| | Local Linear | | Local Polynomial | | Global Polynomial | |
| | <30 km | <50 km | <30 km | <50 km | Full Sample | Full Sample |
| Chongqing | 0.1303 *** | 0.1353 *** | 0.1189 *** | 0.1063 *** | 0.1272 *** | 0.1272 *** |
| | (0.036) | (0.032) | (0.035) | (0.032) | (0.031) | (0.031) |
| Polynomial | linear | linear | quadratic polynomials | quadratic dimensional | cubic dimensional | biquadratic dimensional |
| N | 3223 | 5626 | 3223 | 5626 | 17,435 | 17,435 |
| Adj.$R^2$ | 0.0164 | 0.0181 | 0.0218 | 0.0226 | 0.0225 | 0.0225 |

Note: *** represent 1% levels of significance respectively.

We also examine the relationship between resource misallocation and TFP and find that the greater the degree of resource misallocation, the smaller the TFP, i.e., resource misallocation reduces the TFP of firms, which is consistent with the findings in the established literature [36,37].

**Table 11.** Results of resource misallocation measured using the standard deviation of tax rate.

| | (1) Misallocation | (2) Misallocation | (3) Misallocation | (4) Misallocation | (5) Misallocation | (6) Misallocation |
|---|---|---|---|---|---|---|
| | Local Linear | | Local Polynomial | | Global Polynomial | |
| | <30 km | <50 km | <30 km | <50 km | Full Sample | Full Sample |
| Chongqing | 1.3495 *** | 1.2993 *** | 1.2983 *** | 1.0686 *** | 1.3214 *** | 1.3227 *** |
| | (0.340) | (0.282) | (0.347) | (0.301) | (0.293) | (0.293) |
| Polynomial | linear | linear | quadratic polynomials | quadratic dimensional | cubic dimensional | biquadratic dimensional |
| N | 3223 | 5626 | 3223 | 5626 | 17,435 | 17,435 |
| Adj.R$^2$ | 0.0178 | 0.0215 | 0.0220 | 0.0259 | 0.0231 | 0.0231 |

Note: *** represent 1% levels of significance respectively. All regressions control for two-dimensional geographic variables. Values in parentheses are clustering robust standard errors, all regression models are adjusted for clustering at the county level.

## 7. Conclusions

Under China's current administrative system, political hierarchy measures the amount of political and economic resources that the local government can control, directly affecting the flow and redistribution of resources, and it thus has important implications for firm behavior. This paper empirically examines the long-term effects of city-level upgrading on micro-firm TFP, using Chongqing's upgrading to a provincial-level municipality as an exogenous shock and data on industrial firms in Chongqing and Sichuan in 1995 and 2013. The results show that: (1) there is no significant difference in TFP between firms near the border of Chongqing and Sichuan before Chongqing's separation, but after Chongqing's elevation to a provincial-level municipality, the TFP of Chongqing firms is significantly lower than that of Sichuan provincial firms, by about 5.5–7.4%. (2) In the robustness test, we used the DID regression discontinuity approach, replaced the TFP measure, control for fixed effects, adjusted the bandwidth and boundaries, used placebo tests, and eliminated the effect of firm migration. All test results are supportive of our conclusions. (3) The mechanism analysis finds that after Chongqing's upgrading, the land price in Chongqing is 25% lower than that in Sichuan Province, which means that under the incentive of economic development, Chongqing attracts more low-productivity firms to settle, leading to the "crowding out" of high-productivity firms. At the same time, we find that Chongqing's upgrading exacerbates the degree of resource misallocation of firms through policy intervention. Together, land price and resource misallocation lead to a decline in firms' TFP.

The findings of this paper provide empirical evidence for the debate on the relationship between the government and the market. Under China's unique political system, the elevation of the city level has expanded the government's economic and social management authority, and thus has a significant impact on firms. China has claimed to promote a better combination of an efficient market and a responsive government. Accordingly, this paper suggests that the efficiency of land use should be improved to avoid inefficient government investment; policies such as the establishment of a land inspection system should be explored. Second, government intervention should be appropriately reduced, and ex-ante review and ex-post monitoring of government intervention should be strengthened.

Although the process of administrative elevation is rather exclusive to countries with very strictly top-down political structure such as China, this paper still provides significant insight into the role the government can play in the economic development process. The effect the administrative power can have on the private enterprises is largely dependent on the incentive structures for the ones wielding such power. In a truly representative democracy, because the prosperity of the private enterprises is of great importance to the local residents, thus empowering the local government can be rather beneficial, whereas in an authoritarian government structure, the officials are incentivized by the desires of pleasing their higher-ups who picked them for their jobs, whose priorities do not always align with the that of the local private enterprises. The empirical evidence provided by

this paper can be used against the idea of giving the government more power in economic management, when the representativity of such government is questionable.

**Author Contributions:** Conceptualization, W.H.; methodology, W.H.; software, Y.Y.; validation, W.H. and Y.Y.; formal analysis, W.H.; investigation, W.H.; resources, W.H.; data curation, W.H.; writing—original draft preparation, W.H.; writing—review and editing, Y.Y.; visualization, W.H.; supervision, W.H.; project administration, Y.Y. All authors have read and agreed to the published version of the manuscript.

**Funding:** This research received no external funding.

**Institutional Review Board Statement:** This study uses data already available and involves no ethical questions.

**Informed Consent Statement:** Not applicable.

**Data Availability Statement:** No new data were created.

**Conflicts of Interest:** The authors declare no conflict of interest.

## Appendix A

The triangular kernel is a type of weight function used in kernel density estimation, which is a non-parametric method for estimating the probability density function of a random variable. The triangular kernel is one of the most used kernels in density estimation because it is simple to implement and has some desirable properties.

The triangular kernel is defined as follows:

$$w_i = \max\left(1 - |x_i - c|/h,\ 0\right)$$

where $w_i$ is the weight assigned to observation $i$, $x_i$ is the value of the running variable for observation $i$, $c$ is the cutoff point, and h is the bandwidth, max $(, \ldots )$ is the maximum function that takes the larger of the two values.

To ensure that the weights sum to 1, the weights are normalized by dividing each weight by the sum of all weights:

$$w_i = w_i / \sum_j w_i$$

where $w_i$ is the weight assigned to observation i and $\sum_j w_i$ is the sum of all weights. The kernel function is symmetric around zero, with a maximum value of 1 at x = 0, and decays linearly to zero as $|x|$ increases beyond h.

In our paper, the triangular kernel is used in the context of regression discontinuity designs, where it is used to weight the observations in the estimation of the treatment effect at the discontinuity point. By using the triangular kernel, we can estimate the treatment effect of the RDD more accurately by giving more weight to observations closer to the discontinuity point, while minimizing the influence of observations that are further away. Overall, the triangular kernel is a simple but powerful tool for density estimation and regression discontinuity analysis.

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
