# Peer review of "Political Hierarchy and Long-Term Effects on TFP: Evidence from a Provincial Elevation in China"

_sustainability, doi:10.3390/su15086363_

Round 1
Reviewer 1 Report
Dear authors,
I enjoyed reading this paper. This paper examines the effects of political hierarchy on firms' long-term total factor productivity. Here some comments which I hope you will find them useful:
Please remove the empirical numbers findings from the abstract “We find that before Chongqing's elevation to a provincial-level municipality, the TFP of 12 Chongqing firms was not significantly different from that of Sichuan firms, but by 2013, the TFP of 13 Chongqing firms near the border was significantly lower than that of Sichuan firms by about 5.5-14 7.4%. The mechanism analysis shows that the land transaction price in Chongqing is about 25% 15 lower than that in Sichuan Province, which leads to the entry of low-productivity firms and the 16 "crowding out" of high-productivity firms. We also find that government intervention in Chongqing 17 exacerbates the degree of resource misallocation at the firm level, which together lead to the decline 18 in firm TFP.”
The abstract should discuss the aim, sample, method, finding and implications.
Explain why in table two, the adjected R is very low
What is your control group for the DID regression?
Expand the discussions on implications, limitations and avenues for future research.
Please refer to some relevant literature from the journal and broader literature, such as the below references.
Elmarzouky, M., Hussainey, K., Abdelfattah, T. and Karim, A.E., 2022. Corporate risk disclosure and key audit matters: the egocentric theory. International Journal of Accounting & Information Management.
Shohaieb, D., Elmarzouky, M. and Albitar, K. (2022), "Corporate governance and diversity management: evidence from a disclosure perspective", International Journal of Accounting & Information Management, Vol. 30 No. 4, pp. 502-525. https://doi.org/10.1108/IJAIM-03-2022-0058
Reviewer 2 Report
Novel contribution simply missing. Current article is a literature and data study, but what are the future development recommendations nations, strategies etc. proposed here?
Author Response
This paper is mainly an empirical study, but we understand your concerns. The main question we are trying to answer is the relationship between government’s power in resource allocation and the development of private enterprises.
The increase in the government’s economic power can come from a good incentive, but it often leads to the misallocation of enterprise resources. The main reasons are as follows: Firstly, enterprises seek rent. Because government subsidies are not determined by the market mechanism, enterprises conduct rent-seeking behaviors in order to obtain more preferences and subsidies, that is, enterprises allocate productive inputs to non-productive activities, leading to resource mismatch. Secondly there is the problem of information asymmetry. Government subsidies and preferences may face the problem of adverse selection beforehand, that is to say, in order to obtain more subsidies and preferences, enterprises will release false signals to deceive the government, leading to resource mismatch.
The main limitation of our paper is that it mainly provides empirical cases affecting enterprise TFP from the perspective of government level promotion. Whether increasing the political status of local government has the same effect on other economic factors required further research. At the same time, although our research shows that improving the political status of cities can significantly reduce the TFP of enterprises, what the government cares most when making decisions is often not the TFP of enterprises. In China, GDP statistics have long been regarded as the core or even the only indicator to evaluate government performance. The government officials have stronger motivation to pursue the goal of high GDP, and increasing the size of enterprises has become the core for policy makers. Enterprises are attracted by lower land expropriation cost, so the sacrifice of efficiency became inevitable. This development model has led to a long period of rapid economic growth in China, and directly led to high pollution, high energy consumption, zombie enterprises and other adverse consequences.
Therefore, we must be quite honest that the lesson provided in our paper is hard to apply to other countries such as the United States, but rather specific to the authoritarian top-down structure of the Chinese government.
Reviewer 3 Report
My comments on the paper, Political Hierarchy and Long-Term Effects on TFP: Evidence from a Provincial Elevation in China, submitted to Sustainability.
1. The literature review is especially plodding, each paragraph begins with an author’s name. There needs to be a unifying summary for thissection at the end of the literature review.
2. If the author is making heavy use of a particular software package, they may want to acknowledge this in the text or after each summarizing the empirical results.
3. Also make sure that the literature review is written in the past tense as the work has already been completed; similarly to the empirical estimates.
4. A major problem with this paper is the lack of theory. What is the theory that underlines the empirical estimation of this paper? Once the theory has been identified, the authors can commence the empirical estimation.
4. The acronym TFP is not defined anywhere in the paper. Please define this acronym.
5. Page 3 Line 142, should "exogenous" be replaced with "sudden?"
6. Page 3 line 121 before could the authors should insert the word and
7. Page 4 line 172 Many literatures should be replaced with In the past literature
8. On page 4 line 178 the authors mentioned empirical tests. What type of empirical tests?
9. Page 8, lines 289-290, how does this approach of the spatial standard error differ from other spatial standard errors?
10. Throughout the empirical tables, are the standard errors robust? Clustered? Using the Cohen spatial standard error?
11. The authors mentioned the double difference-in differences (DID). The authors want to discuss under the methodology what the double difference-in differences (DID) are for the readers who may not be familiar with this empirical method. Also, the authors may want to explain more about the spatial regression discontinuity approach for readers not familiar with this approach
12. How was the weight matrix developed for the spatial regression discontinuity approach? The use of the weight matrix is crucial in empirical spatial regressions.
13. The references in the paper do not conform to the guidelines of Sustainability.
14. Could the analysis presented in this paper be generalizable to other types of areas?
Round 2
Reviewer 2 Report
Please describe within the conclusion and outlook parts how your approach, or the studied phenomenon and its positive development in future, would bring a significant transition in the situation described. I understand the novelty of your study approach, but please describe more clear the novelty resulting from applying it, in real life.
Thank you in advance!
Author Response
Thank you very much for your patience and time. We have added the following paragraph in our conclusion section according to your advice.
"Although the process of administrative elevation is rather exclusive to countries with very strictly top-down political structure such as China, this paper still provides significant insight into the role the government can play in the economic development process. The effect the administrative power can have on the private enterprises is largely dependent on the incentive structures for the ones wielding such power. In a truly representative democracy, because the prosperity of the private enterprises is of great importance to the local residents, thus empowering the local government can be rather beneficial, whereas in an authoritarian government structure, the officials are incentivized by the desires of pleasing their higher-ups who picked them for their jobs, whose priorities do not always align with the that of the local private enterprises. The empirical evidence provided by this paper can be used against the idea of giving the government more power in economic management, when the representativity of such government is questionable."
Reviewer 3 Report
My comments on the second review of the paper, Political Hierarchy and Long-Term Effects on TFP: Evidence from a Provincial Elevation in China submitted to Sustainability.
1. The references still do not conform to the guidelines of Sustainability. The authors need to make sure that these references do conform before publication.
2. Values in brackets are clustering robust 548 standard errors, all regression models are adjusted for clustering at the county level. I do not see the brackets. Did the authors mean parentheses?
3. In the following sentences in lines 561-563, In the robustness test, we further use the DID regression discontinuity approach, replace the TFP measure, control for fixed effects, adjust the bandwidth and boundaries, use placebo tests, and eliminate the effect of firm migration. All test results are very supportive of our conclusions. It should be rewritten as In the robustness test, we used the DID regression discontinuity approach, replaced the TFP measure, control for fixed effects, adjusted the bandwidth and boundaries, used placebo tests, and eliminated the effect of firm migration. All test results are supportive of our conclusions.
4. The review of the literature should be written in past tense as the work has already been completed; similarly with the methods and the empirical results. There is a mixture of tenses throughout the paper.
5. In lines 346 and 347, the following sentence needs to be rewritten: By controlling the fixed time effect, we can capture the impact of macro-impact from the national level on the results, so as to better identify the long-term impact of the city-level promotion. This sentence should be rewritten as y controlling the fixed time effects, we captured the impact of macro-impact from the national level on the results, which better identifies the long-term impact of the city-level promotion.
6. It is not still clear how the weight matrix was developed. Please elaborate how this weight matrix was developed.
Author Response
- The references have been revised as required.
- The world 'brackets' have been replaced by 'parentheses'.
- The sentence has been revised according to your advice.
- The review of literature has been revised as required.
- The sentence has been revised according to your advice.
- We have included an appendix to explain the way the matrix was developed, it is the Appendix A at the end of the manuscript.
Round 3
Reviewer 3 Report
My comments on the third review of the paper, Political Hierarchy and Long-Term Effects on TFP: Evidence from a Provincial Elevation in China submitted to Sustainability.
1. Line 294, the sentence The way the weight matrix is constructed is explained in appendix A. should be rewritten as The construction of the weighting matrix is provided in Appendix A.
2. Line 610 Where should be where